# Single-Cell and Bulk RNA Sequencing Reveal SPINK1 and TIMP1 as Epithelial Cell Marker Genes Linked to Colorectal Cancer Survival and Tumor Immune Microenvironment Profiles

**DOI:** 10.3390/ijms262411964

**Published:** 2025-12-11

**Authors:** Noor N. Al-Bzour, Zaid Nassar Abu-Rjai’, Ayah N. Al-Bzour, Abdulrahman Qasaymeh, Anwaar Saeed, Azhar Saeed

**Affiliations:** 1Department of Medicine, Jordan University of Science and Technology, Irbid 22110, Jordan; 2Department of Medicine, Yarmouk University, Irbid 21163, Jordan; 3Department of Medicine, Division of Hematology and Oncology, University of Pittsburgh Medical Center (UPMC), Pittsburgh, PA 15219, USA; 4UPMC Hillman Cancer Center, Pittsburgh, PA 15219, USA; 5Department of Pathology and Laboratory Medicine, University of Vermont Medical Center, Burlington, VT 05401, USA

**Keywords:** colorectal cancer, epithelial cells biomarkers, survival, single-cell analysis, tumor immune microenvironment

## Abstract

Colorectal cancer (CRC) is a major cause of cancer death, with the tumor microenvironment and gene expression influencing outcomes. Identifying survival-associated epithelial marker genes (EMGs) may improve prognosis and guide therapy. We obtained single-cell RNA-sequencing (scRNA-seq) data from CRC patients (*n* = 23,176 cells) from the TISCH database to identify EMGs through differential expression analysis. These were intersected with malignant cell markers. We used bulk RNA-seq data from TCGA-COAD (*n* = 375) to assess EMG prognostic value via univariable Cox analysis, followed by LASSO regression. Significant genes were evaluated using multivariable Cox models. An EMGs-based risk score was developed and validated using GSE39582 (*n* = 585) and GSE17536 (*n* = 177). Immune infiltration was assessed using xCELL and TIMER algorithms. A total of 107 EMGs were identified and assessed in TCGA data. Cox analysis identified 18 survival-related EMGs, which were narrowed by LASSO to SPINK1 and TIMP1. Multivariable analysis confirmed SPINK1 (HR: 0.88, 95% CI: 0.79–0.97, *p* = 0.009) and TIMP1 (HR: 1.66, 95% CI: 1.29–2.13, *p* < 0.001) as independent survival predictors. Patients were classified into high- (*n* = 187) and low-risk (*n* = 188) groups. The low-risk group had significantly better overall and disease-free survival. Immune profiling revealed distinct patterns, where the high-risk group showed higher dendritic cells, memory T-cells, macrophages, and immune checkpoint expression, while the low-risk group showed enrichment of NK cells, plasma cells, and CD4+ T-helper cells. These findings were validated in the GSE39582 and GSE17536 cohorts. EMGs have prognostic value in CRC, with SPINK1 and TIMP1 as independent survival predictors. Distinct immune patterns support integrating EMGs with immune profiling for improved risk stratification and personalized treatment.

## 1. Introduction

Colorectal cancer (CRC) is the third most diagnosed cancer and the second leading cause of cancer-related deaths worldwide [1]. CRC risk is strongly influenced by age, with the likelihood significantly increasing after the age of 50. Early-onset cases are rare and often linked to inherited conditions. Other risk factors include a personal history of CRC, inflammatory bowel disease, and a family history of CRC—particularly in relatives diagnosed before age 50 [2]. A critical understanding of the heterogeneous nature of CRC has been uncovered due to histopathological, genetic, immunological, and molecular analyses [3].

Early-stage CRC, localized disease (Stages I and II), has 5-year survival rates that can reach 90%. While late-stage CRC, which is attributed to metastasis, has a 13.1% survival rate [4].

Epithelial–Mesenchymal transition (EMT) is recognized as one of the factors regulating metastasis, along with the tumor microenvironment [5]. EMT causes significant shape changes in cells, as they lose many of their epithelial characteristics and gain a mesenchymal-like phenotype [6].

Single-cell RNA sequencing (scRNA-seq) allows for the analysis of heterogeneous tumors and the identification of cell compositions at a deep resolution in the tumor microenvironment. In contrast, bulk RNA sequencing involves sequencing a mixture of millions of cells simultaneously, which obscures the properties of individual cells [7]. Therefore, we applied scRNA analysis to uncover potential cellular heterogeneity and diversity within what would otherwise appear as homogenous epithelial cells. By identifying epithelial cell marker genes, we then linked these findings with bulk RNA sequencing data to assess survival outcomes.

Several studies investigated potential epithelial cell markers as potential drug targets for CRC. In our work, we aim to provide a better understanding of specific epithelial cell markers associated with survival in CRC and investigate their interaction with the tumor immune microenvironment, thereby expanding the existing clinical understanding. This would ultimately help in improving patient stratification, follow-up, and treatment options.

In this study, we investigated CRC at single-cell resolution to identify epithelial cell marker genes and correlated these findings with bulk RNA data to assess their association with survival outcomes. Identifying such prognostic factors will aid in developing more precise therapeutic targets for cancer, potentially reducing toxicity. The overall study flowchart is shown in Figure 1.

## 2. Results

### 2.1. Identification of EMGs

A total of 23,176 cells from 7 CRC tissues were isolated into 20 distinct cell clusters through unsupervised clustering (Figure 2A). These were subsequently annotated into 12 cell types by grouping clusters with shared lineage identity based on marker genes provided by the TISCH pipeline. Following cell annotation and differential expression analysis between cell clusters, 12 cell clusters were identified based on cell-specific markers (Figure 2B). We selected the epithelial cell cluster, which represents normal epithelial tissue cells. DEGs from each cell cluster were identified using the TISCH database, and a total of 160 significant EMGs were identified, with 107 markers associated with malignant cells (Appendix A). The overlap between these EMGs and the gene expression data obtained from TCGA-CRC yielded a total of 92 EMGs. Figure 2C shows the interaction heatmap and network visualization that were generated from pre-computed cell–cell communication analysis available in the TISCH database. TISCH integrates the CellChat algorithm to infer ligand–receptor-based signaling between cell clusters, and CellChat quantifies the number of significant ligand–receptor pairs between any two cell populations using its statistical framework, and the number of significant interactions is used as the interaction score.

### 2.2. Establishment of EMG Prognostic Gene Signature

Univariable Cox regression analysis was conducted on the 92 EMGs with age, sex, and tumor stage to identify the survival-associated EMGs using the TCGA-CRC cohort. Based on the criterion of *p*-value < 0.05, a total of 18 genes in addition to age were identified and fitted in the LASSO-Cox regression model with 10-fold cross-validation (Figure 3A,B) based on the optimum λ value. Of these, 3 variables were selected as significant predictors (*p*-value < 0.05), including SPINK1, TIMP1, and age. Sex and tumor stage were not statistically significant in the univariable analysis. Next, we performed a multivariable Cox regression analysis to identify independent predictors of survival, and SPINK1, TIMP1, and age showed significant associations (Figure 3C). SPINK1 exhibited an HR < 1 (95% CI: 0.79–0.97), indicating that it is associated with better outcomes, whereas gene TIMP1 showed an HR > 1 (95% CI: 1.29–2.13), indicating its association with poor survival outcomes. Increasing age was associated with poor outcomes as well, with HR > 1 (95% CI: 1.01–1.04).

The risk score for each patient was calculated using the following formula:Risk score = (−0.128 × SPINK1 expression) + (0.506 × TIMP1 expression) + (0.029 × Age)

Based on the median risk score, patients were divided into low-risk and high-risk groups. Survival analysis was carried out between the two groups and visualized using the Kaplan–Meier curve and log-rank test (Figure 3D,E). The low-risk group exhibited significantly better OS and DFS (*p*-value: <0.0001, 0.0072, respectively).

In addition, we assessed the survival probability between the groups at 1-year, 3-year, and 5-year, which were (86%, 67%, and 48%, respectively) in the high-risk group, and (96%, 90%, and 77%, respectively) in the low-risk group (*p*-value < 0.001).

Complementary analyses using the continuous risk score confirmed its prognostic value, with a time-dependent ROC AUC (68.5–68.9%) at 1–5 years, a C-index of 0.687, and restricted mean survival time (RMST) analysis showing significantly shorter survival in the high-risk group (RMST difference −28.3 months, 95% CI: −46.4 to −10.3, *p* = 0.002).

Boruta algorithm results identified six informative EMGs, which include: CD24, CD37, CD3D, SPINK1, TFF1, and TIMP1. Importantly, TIMP1 and SPINK1, the two genes identified by our original LASSO-based approach, were consistently selected by Boruta, supporting their robustness as prognostic biomarkers.

We compared the clinicopathological features between the two groups (Table 1). Significant association was observed in the vascular invasion indicator and the lymphovascular invasion indicator, which were both higher in the high-risk group. Additionally, patients exhibited a significant association with the primary site of the tumor. In the high-risk group, 55% of the patients had their tumor in the right colon, compared to 38% in the low-risk group. In contrast, 45% of the high-risk group had their tumors on the left side compared to 62% in the low-risk group (*p*-value: 0.002).

### 2.3. Validation of the Risk Model

GSE39582 (*n* = 585) and GSE17536 (*n* = 177) were used as validation cohorts to assess the robustness of the risk model. The risk score for each patient was calculated as described above, and patients were grouped into high- and low-risk groups based on the median risk score. Similarly to the TCGA-CRC results, the low-risk group showed significantly better overall survival in both cohorts (Figure 4). Similarly, we assessed the 1-year, 3-year, and 5-year survival outcomes. In the GSE39582 cohort, the high-risk group showed survival rates of 92%, 74%, and 64%, respectively, compared with 95%, 81%, and 71% in the low-risk group (*p* = 0.031). In the GSE17536 cohort, the high-risk group showed 1-year, 3-year, and 5-year survival rates of 83%, 63%, and 44%, whereas the low-risk group showed 93%, 77%, and 69% (*p* = 0.003).

### 2.4. Immune Infiltration and Modulation

The expression of the SPINK1 gene was found to negatively correlate with the infiltration of CD8+ T cells, myeloid dendritic cells, and neutrophils, with these associations reaching statistical significance according to the TIMER algorithm. In contrast, the negative correlations between SPINK1 expression and the infiltration of B cells and CD4+ T cells were not statistically significant. The expression of the TIMP1 gene demonstrated a significant positive correlation with the infiltration of CD8+ T cells, dendritic cells, macrophages, and neutrophils, a positive but not significant correlation with CD4+ T cells, and a significant negative correlation with B cells (Figure 5A,B).

Using the xCELL algorithm to compare immune and stromal cell fractions, as well as microenvironment scores, between high-risk and low-risk patient groups (Table 2), the findings revealed key differences in immune composition and microenvironment features, with significant implications for the underlying biology of the risk groups. Regarding the immune cell population, the activated dendritic cells (aDC), conventional dendritic cells (cDC), and immature dendritic cells (iDC) were significantly higher in the high-risk group compared to the low-risk group. These findings may reflect an enhanced immune activation state in high-risk patients, potentially indicating increased antigen presentation activity. Higher levels of macrophages, including both M1 (pro-inflammatory) and M2 (anti-inflammatory) subtypes, were observed in the high-risk group. This dual presence may signify a complex interplay of pro- and anti-tumor immune responses in these patients. On the other hand, CD4+ memory T-cells, Th1, and Th2 cells were significantly more abundant in the low-risk group, indicating enhanced adaptive immunity in this population. NK cells were also significantly elevated in the low-risk group, suggesting enhanced innate anti-tumor immunity.

Regarding stromal cells, the high-risk group demonstrated markedly higher levels of fibroblasts and endothelial cells, indicating an enriched tumor stroma and angiogenesis. Mesenchymal Stem Cells (MSCs) were significantly elevated in the high-risk group, aligning with their known roles in promoting tumor progression and immunosuppression. Moreover, the tumor microenvironment score, immune score, and stromal score were significantly higher in the high-risk group (*p*-value < 0.001).

In the high-risk group, GSVA showed significant upregulation of hallmark pathways associated with angiogenesis, epithelial–mesenchymal transition, myogenesis, inflammatory response, and key signaling pathways, including KRAS, IL6-JAK-STAT3, and TNFα via NF-κB.

Conversely, the upregulated pathways in the low-risk group were primarily linked to cell cycle progression, DNA repair, oxidative phosphorylation, MYC targets, and lipid metabolism (Figure 6A).

The Reactome pathway analysis showed upregulation of tRNA processing in the mitochondria in the low-risk score group. The high-risk score group showed upregulation of pathways such as GLI-mediated transcriptional activation in Hedgehog signaling, gap junction transmission, ECM interactions, collagen biosynthesis, and various defective glycosaminoglycan biosynthesis-related pathways (e.g., CHSY1, CHST6, CHST3, CHST14, B4GALT1) (Figure 6B).

There was a significantly higher expression of immune checkpoints (PD-1, PDL-1, CTLA-4, and LAG3) in the high-risk group. To explore the potential presence of exhausted CD8+ T-cells, we stratified patients based on patterns of co-expression of the following immune checkpoints: High PD1 + High LAG3, High PD1 + High CTLA-4, and High PD1 + High PDL-1. Patients meeting these criteria were considered to have higher likelihood of CD8+ T-cell exhaustion. The high-risk group showed a significantly higher frequency of patients (62%) meeting these criteria, in comparison to the low-risk group (31%, *p*-value < 0.001), as shown in Table 3. In support of this observation, additional analysis showed significantly higher expression levels of T-cell exhaustion markers, EOMES and TIGIT, in the high-risk group (*p*-values < 0.001, Table 3).

## 3. Discussion

EMGs expressions are identified as an important prognostic factor for CRC. Studies have found that EMGs are correlated with the overall survival of CRC patients and show a significantly shorter survival time [8].

Serine protease inhibitor Kazal type 1 (SPINK1) is a trypsin kinase inhibitor that has been linked to inflammation, cancer cell proliferation, and carcinogenesis [9]. It is predominantly produced by pancreatic acinar cells. However, it is widely expressed in extra-pancreatic tissues, particularly in the gastrointestinal tract, indicating additional biological roles, including protection of the colonic and gastric mucosa [10,11,12]. Historically, Higashiyama et al. and Tomita et al. showed the production of SPINK1 by colon cancer tissue [13,14]. Previous studies showed that SPINK1 and trypsin are co-expressed in CRC, suggesting a potential protective role for SPINK1 against tumor invasion [15]. A study by Rasanen et al. reported that elevated expression of SPINK1 is associated with poor outcomes in CRC and showed a positive correlation between SPINK1 and interleukin-6 (IL-6), indicating its role as an acute-phase reactant. Their results reveal a novel connection between inflammatory signals from the tumor microenvironment and elevated SPINK1 levels. These findings suggest potential therapeutic implications for targeted therapy as they confirm SPINK1’s role as an acute-phase reactant and its involvement in paracrine crosstalk within the colon cancer tumor microenvironment [16].

Tissue inhibitor matrix metalloproteinase 1 (TIMP1) is a matrix metalloproteinase (MMP) that regulates the interaction with the extracellular matrix (ECM), such as cell growth, tumor cell invasion, and metastasis. As a natural regulator of MMP, TIMP1 plays a role beyond merely inhibiting MMP activity; it also influences immune responses, angiogenesis, and tumor cell migration within the tumor microenvironment [17]. Studies suggest that TIMP1 levels have been linked to the prognosis of several cancer types, including breast and pancreatic cancer. Moreover, its expression is consistently upregulated during malignant transformation and negatively correlates with CRC prognosis [18].

Our study showed two EMGs that are associated with survival in CRC, including the SPINK1 gene, which is associated with better outcomes, and the TIMP1 gene, which is associated with worse outcomes. In this context, studies have shown that SPINK1 has been identified as an independent prognostic factor in CRC and that high SPINK1 expression correlates with a favorable prognosis [19]. Additionally, the SPINK1 gene is linked to a favorable prognosis in CRC patients, as reported in a study investigating its functional role in cancer. Animal studies using murine models treated with SPINK1 demonstrated upregulation of pathways such as PI3K/AKT and MAPK/ERK. These findings suggest that patients with higher SPINK1 expression may benefit from targeted therapies against the AKT/MAPK pathway [20]. Functionally, the expression of SPINK1 is upregulated by inflammatory signals from the tumor microenvironment, particularly interleukin-6 (IL-6) Via the STAT3 pathway [16]. In contrast to previous findings where high SPINK1 expression was associated with poor prognosis in CRC, the results of Chen et al. demonstrate that high SPINK1 expression correlates with improved overall survival in stage IV KRAS wild-type CRC patients receiving cetuximab-based anti-EGFR therapy. This suggests that SPINK1 may function not only as a prognostic marker but also as a predictive biomarker for response to EGFR-targeted therapy. It is known that SPINK1 activates key EGFR downstream pathways, including PI3K/AKT and MAPK/ERK, indicating that SPINK1-high tumors may be more dependent on EGFR signaling for proliferation and therefore more susceptible to EGFR inhibition. Thus, SPINK1 expression may help stratify patients who are most likely to respond favorably to anti-EGFR therapy, supporting its clinical value in personalized treatment strategies for CRC [9].

Studies on the TIMP1 gene have shown that it is associated with poor prognostic outcomes by promoting cell proliferation and invasion [21]. Additionally, TIMP1 is linked to unfavorable clinicopathological factors, indicating a poor prognosis in CRC patients. High TIMP1 expression correlates with worse prognosis and shorter overall survival compared to low TIMP1 expression, highlighting its significant prognostic value [22]. Similarly, Shen et al. identified TIMP1 as a stemness-related prognostic gene through univariable Cox regression analysis. Their Kaplan–Meier analysis demonstrated that higher TIMP1 expression correlates with lower patient survival rates. Additionally, TIMP1 plays a crucial role in CRC tumorigenesis and metastasis, highlighting its potential as a prognostic indicator. Furthermore, Shen et al. reported that TIMP1 serves as an independent diagnostic marker for CRC, with its presence in platelets contributing to CRC progression [23].

To validate our findings further, the Boruta algorithm confirmed that both TIMP1 and SPINK1 are informative prognostic markers associated with survival outcomes. In addition to these genes, the algorithm identified other candidate genes, including CD24, CD37, CD3D, and TFF1. These discrepancies may arise from the differences in the methodology between LASSO and Boruta. Nevertheless, the consistent identification of SPINK1 and TIMP1 supports our findings.

Compared with a study by Zheng et al. [24], which identified a 9-gene signature based solely on bulk RNA data, our study offers a comprehensive biological resolution by integrating both single-cell and bulk transcriptomic modalities. Through this approach, we dissected the epithelial cell cluster to provide mechanistic insights into the cellular origin of the signature genes.

Our study highlighted a correlation between different immune cells and EMGs-based risk scores. In particular, dendritic cells appeared to play a key role in CRC prognosis, as the high-risk group showed high infiltration of aDC, cDC, and iDC compared to the low-risk group. While dendritic cells play an important role in CRC progression as they interfere with the innate and adaptive immunity and in the establishment and persistence of cancer-induced immunosuppression [25], tolerogenic dendritic cells have been reported to induce T-cell dysfunction and exhaustion; therefore, the functional state of DCs should be further characterized in follow-up studies [26].

In addition, the high-risk group showed increased infiltration of macrophages (M1, M2), and high TIMP1 expression showed a high correlation with M1and M2 infiltration [22]. The shift from M1 macrophages, which promote immune activation, to M2 macrophages, which suppress immune responses, is a defining feature of malignant tumors. While M1 macrophages play an anti-tumorigenic role, M2 macrophages are involved in supporting tumor growth [27,28]. Both subtypes are important for prognosis and can influence the effectiveness of immunotherapy. In a study by Huang et al., immune cell clusters exhibited activation of immune-inflammatory pathways, with a gradual increase in M2 macrophages observed as tumors progressed. The researchers suggest that genes associated with the M2 macrophage trajectory may serve as valuable prognostic indicators and contribute to the variability in clinical outcomes, providing a strong basis for the development of a molecular classification system [29].

Elevation of MSCs was significantly noticed in the high-risk group, aligning with their known roles in promoting tumor progression and immunosuppression. MSCs are implicated in the growth, invasion, and metastasis of colorectal cancer cells [30]. MSCs encourage the progression of CRC Via AMPK/mTOR-mediated NF-κB activation [30]. Conversely, MSCs have been proven to inhibit the progression of CRC by inhibiting chronic inflammation and regulating the intestinal microbiota imbalance [31,32]. This indicates that MSCs can be used as a therapy to treat CRC [33]. Some studies suggest an Epithelial–Mesenchymal transition process, in which epithelial cells acquire a mesenchymal phenotype, impacting CRC prognosis and survival [34].

Our study showed a positive correlation between TIMP1 expression and CD8+ T-cells. Zheng et al. reported that CD8+ T cells play a critical role in the immune response against cancer cells, including CRC. However, its effectiveness in CRC is limited by the tumor microenvironment (TME), where CD8+ T cells can become exhausted or dysfunctional. CD8+ T cells can kill tumor cells by recognizing MHC class I molecules on tumor cells. CRC cells can evade this recognition through downregulating MHC class I molecules or upregulating immune checkpoints, hindering CD8+ T cell antitumor activity. Despite this, unconventional T cells, which do not rely on MHC class I recognition, may help bridge the gap and offer potential therapies like CAR-T cell treatments [35]. While the immune infiltration analysis in our study showed a higher infiltration of CD8+ T-cells in the high-risk group, further analysis revealed that these patients might have higher levels of exhausted CD8+ T-cells due to chronic activation, rendering the poorer prognosis in the high-risk group and inability to mount an effective anti-tumor immune response. Altogether, the immune microenvironment in the high-risk group shows indications of higher exhausted CD8+ T-cells, dendritic cells, and M1/2 macrophages, in addition to the higher infiltration of stromal contents such as fibroblasts and MSCs, suggesting an immunologically active but dysfunctional tumor microenvironment (inflamed but exhausted), which may explain the poor prognosis seen in this group. The findings support the use of immunotherapy-based regimens in high-risk patients while taking into account the suppressive stromal environment that might blunt the immunotherapy response.

Our findings show that NK cells were enhanced in the low-risk group, indicating their innate anti-tumor effects. This may explain the favorable prognosis observed in the low-risk group, attributed to the presence of potent innate immune effectors. NK cells and gamma-delta T-cells are known for their ability to recognize and eliminate tumor cells in an antigen-independent manner, contributing to rapid immune surveillance [36]. NK cells, through cytotoxic granules and IFN-γ production, may help suppress early tumor growth and promote Th1 responses. Likewise, gamma-delta T-cells serve as a critical bridge between innate and adaptive immunity by producing pro-inflammatory cytokines and engaging with antigen-presenting cells. These innate immune features are likely to establish a tumor microenvironment conducive to effective immunosurveillance, thus limiting progression and supporting improved clinical outcomes in the low-risk cohort [37,38].

One limitation of our study is that the prognostic model relies on single-gene features (SPINK1 and TIMP1), whereas gene-pair–based approaches have been reported to offer greater robustness and predictive accuracy by reducing batch effects and normalization biases, suggesting the need for future comparison or integration of gene-pair frameworks [39]. In addition, TIMER and xCell algorithms were applied without tumor purity correction; as a result, the associations derived from these deconvolution methods should be interpreted with caution. Single-cell or purity-adjusted deconvolution will be required to validate immune associations in future studies.

## 4. Materials and Methods

### 4.1. Single-Cell RNA Sequence Data

The scRNA-seq data for CRC patients were downloaded from Array Express E-MTAB-8107 [40], which was deposited in the public Tumor Immune Single-cell Hub (TISCH) database (http://tisch.comp-genomics.org/home/, accessed on 2 November 2024) (version 2.0). The dataset consists of 23,176 cells from 7 CRC tissues, and the scRNA-seq was conducted by the platform of 10× Genomics. The single-cell expression matrix values are normalized using the ‘NormalizeData’ function in “Seurat”, which scales the raw UMI counts in each cell to 10,000. A standardized analysis pipeline, MAESTRO, was applied to each dataset for quality control, clustering, and cell-type annotation. MAESTRO performs systematic quality control (including doublet and low-quality cell removal), normalization, clustering, and curated cell-type annotation before any differential expression analysis is generated. TISCH annotates cells using a combination of three methods, first by incorporating annotations provided in original studies. If available, the expression of the markers of the malignant cells from established research is checked. Lastly, using “InferCNV”, the malignancy status based on the copy number variation findings can be predicted, which ultimately differentiates between malignant and non-malignant cells. For normal cells, the cells are annotated in the MAESTRO method using the differentially expressed genes between cell clusters, followed by manual correction of the cell types according to the annotations provided in original studies [41,42].

### 4.2. Bulk RNA Sequencing Data

We downloaded the gene transcriptome data from The Cancer Genome Atlas (TCGA), for patients with CRC. After removing the patients with incomplete survival outcomes, we included 375 patients in the downstream analysis. Additionally, we downloaded the GSE39582 and GSE17536 CRC dataset from Gene Expression Omnibus (GEO) for validation of the risk model.

### 4.3. Identification of Epithelial Cell Marker Genes (EMGs)

Using scRNA-seq data from TISCH, we identified the differential genes for each cell cluster. Epithelial Cell Marker Genes (EMGs) were filtered using |Log_2_FC| > 0.5 and adjusted *p*-value < 0.05. Additionally, we identified the markers of the malignant cell cluster, and these were crossed with the EMGs to include the EMGs associated with malignancy. The resulting EMGs were further crossed with the gene expression data from TCGA to identify the overlapping genes for further downstream analysis.

### 4.4. Risk Model Development

The TCGA-CRC cohort was used as a training group to identify the EMG-survival-associated genes. A Cox risk model, based on EMGs, was constructed with gene expression data from the training group. To identify genes with significant prognostic value, EMGs, age, sex, and tumor stage were analyzed using univariable Cox regression according to overall survival (OS), and those with *p* < 0.05 were selected. To reduce the risk of overfitting and multicollinearity, we applied the Least Absolute Shrinkage and Selection Operator (LASSO) regression to assess the candidate prognostic genes. The final list of LASSO-selected genes was identified based on the optimal penalty parameter, determined when the LASSO 10-fold cross-validation error was minimized. These LASSO-selected prognostic genes were then fitted to a multivariable Cox regression analysis to establish the final risk model.

Each patient’s risk score was calculated using the following formula:Risk score = βmRNA1 × ExpressionmRNA1 + βmRNA2 × ExpressionmRNA2 + … + βmRNAn × ExpressionmRNAn
where β represents the gene coefficient obtained from the Cox hazard regression model.

Patients were then grouped based on the median risk score into high-risk and low-risk groups. Survival outcomes, including overall survival (OS) and disease-free survival (DFS), were compared between the two groups using a log-rank test and visualized using the Kaplan–Meier curve.

To further confirm the robustness of our feature selection pipeline, we implemented the Boruta algorithm on the EMGs (R package version 9.0.0). This algorithm is a random forest-based wrapper method that accounts for potential gene-gene interactions and identifies informative features while eliminating irrelevant ones.

We further assessed the clinicopathological variables across the two risk groups using the Pearson Chi-Squared test, Wilcoxon rank sum test, and Fisher’s exact test.

### 4.5. Immune Infiltration Analysis

Using the TIMER (web server, TIMER 2.0) [43], we explored the tumor immune microenvironment and immune infiltration across six different immune cells, including CD8+ T cells, CD4+ T cells, B cells, Macrophages, Dendritic cells, and Neutrophils. Additionally, we conducted an immune infiltration analysis using the xCell algorithm (R package version 1.0.0) to identify the infiltrated immune and stromal cells between the risk groups and assess the risk score capacity for immune infiltration prediction. Additionally, we used Gene Set Variation Analysis (GSVA) to explore and compare the tumor microenvironment characteristics between the risk score groups [44]. GSVA was performed using the ‘GSVA’ R package (version 1.48.2) with default parameters. Hallmark and Reactome gene sets were obtained from MSigDB. *p*-values were adjusted using the Benjamini–Hochberg method (*p*-value < 0.05).

### 4.6. Statistical Analysis

Descriptive statistical analysis was conducted on the clinicopathological data of the CRC patients. Continuous variables were reported as mean ± standard deviation, while categorical variables were presented as frequencies and proportions. The Wilcoxon rank-sum test was applied to ordinal variables, and Fisher’s exact test and Pearson’s chi-square test were used for categorical variables across the groups. Features with a *p*-value < 0.05 were considered statistically significant. All data processing and analysis were performed using R software (version 4.4.1).

## 5. Conclusions

In conclusion, our results showed a significant role of EMGs in CRC prognosis. Among them, SPINK1 and TIMP1 emerged as independent predictors of survival. Differential immune enrichment patterns between risk groups show the potential of integrating EMGs with immune landscape analysis to refine risk stratification and inform personalized therapeutic strategies.

## Figures and Tables

**Figure 1 ijms-26-11964-f001:**
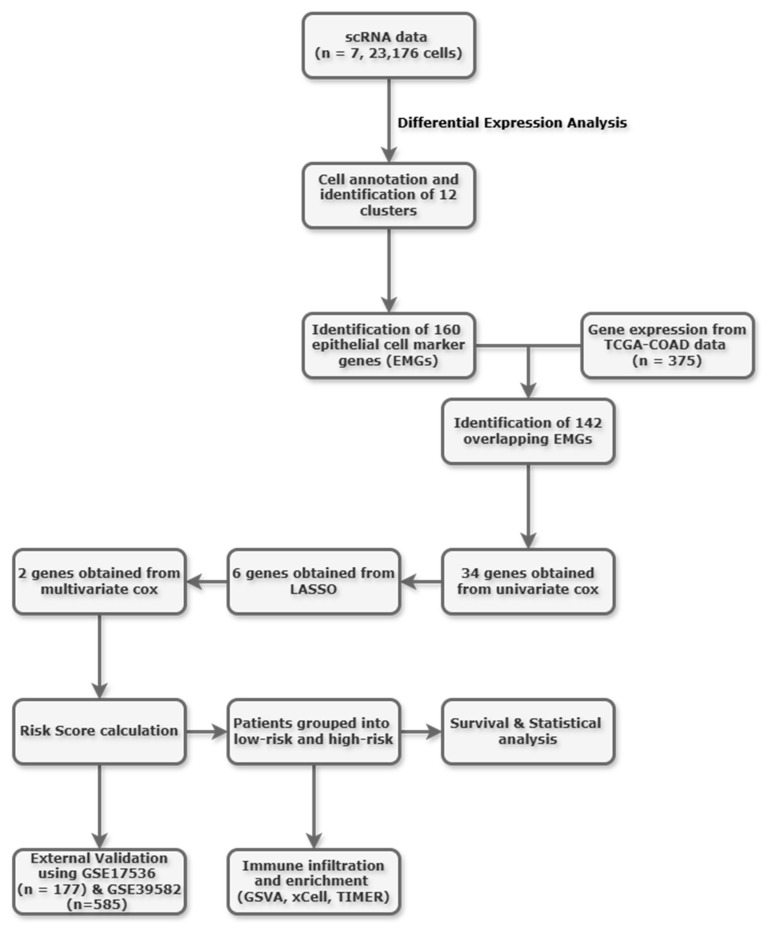
Study workflow.

**Figure 2 ijms-26-11964-f002:**
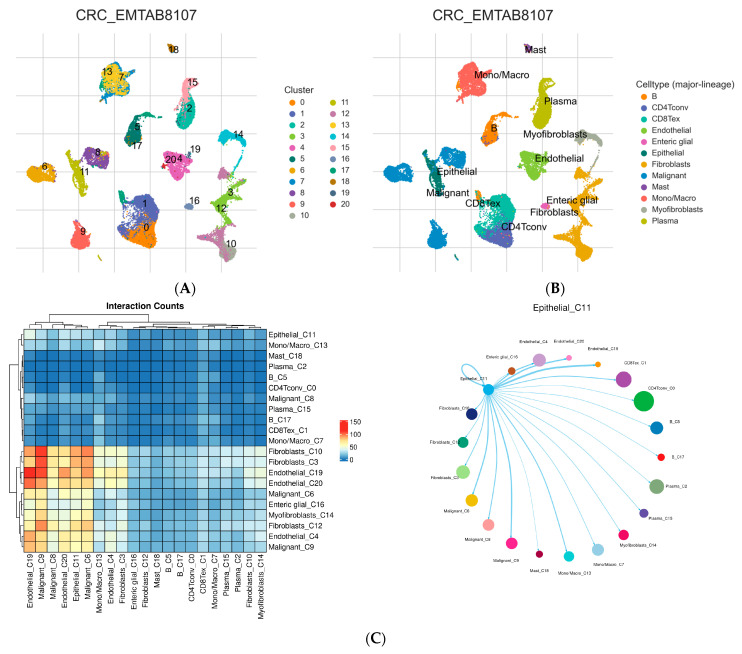
Identification of epithelial cell marker genes using scRNA-seq. (**A**) The UMAP plot of 23,176 cells from primary CRC tissues. (**B**) UMAP plot of the 12 cell clusters. (**C**) Interaction heatmap showing the total counts of significant ligand–receptor interactions between each cluster pair, and the network diagram showing the interaction partners of a selected cluster with edge widths proportional to interaction counts.

**Figure 3 ijms-26-11964-f003:**
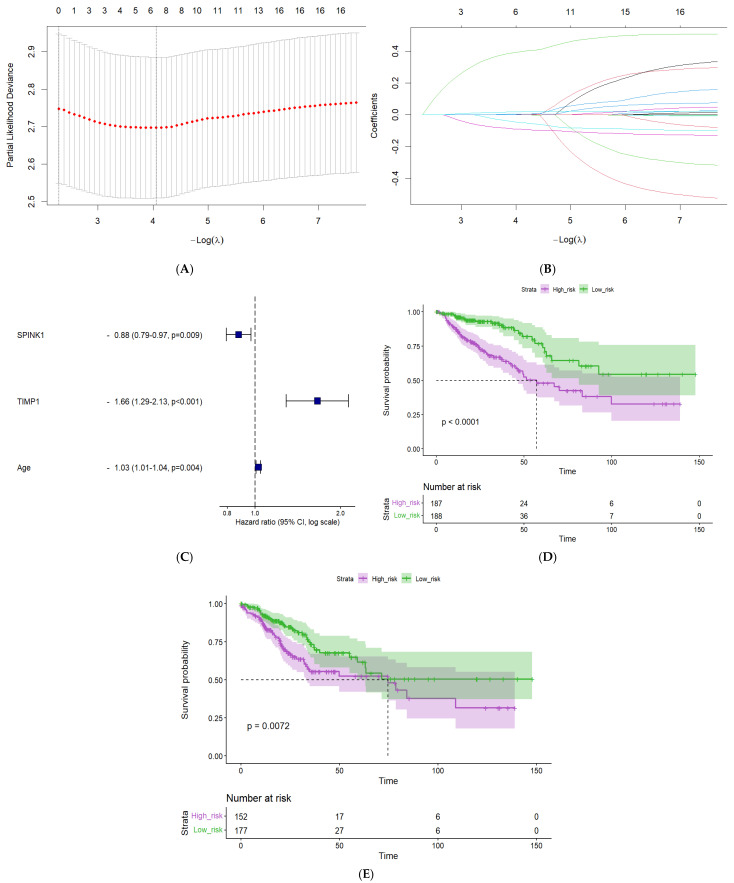
Risk score model development. (**A**) LASSO regression of the 18 genes & age. (**B**) Tuning parameter (λ) selection cross-validation curve of these genes. (**C**) Forest plot showing the 3 variables included in the multivariable Cox analysis. (**D**) Kaplan–Meier curve of OS stratified by the two risk groups. (**E**) Kaplan–Meier curve of DFS stratified by the two risk groups.

**Figure 4 ijms-26-11964-f004:**
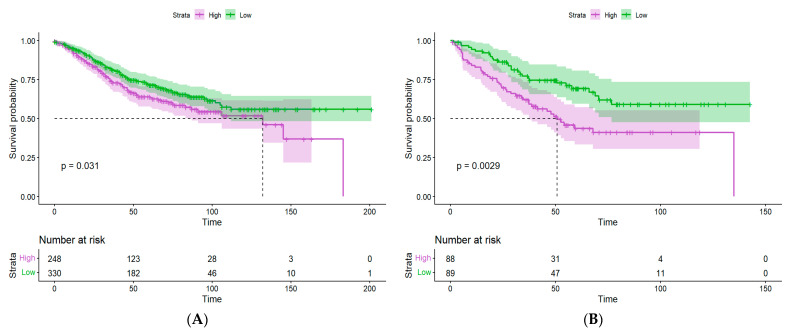
Validation of the risk score in the GEO CRC cohorts. (**A**) Kaplan–Meier curve of the OS stratified by the patients in the two risk groups in GSE39582. (**B**) Kaplan–Meier curve of the OS stratified by the patients in the two risk groups in GSE17536.

**Figure 5 ijms-26-11964-f005:**
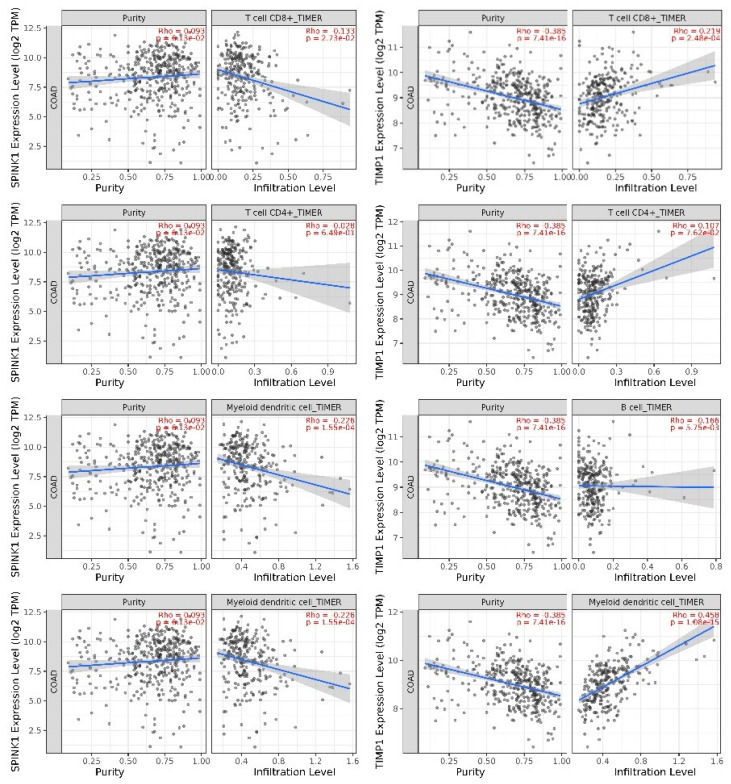
Immune cell infiltration associated with the two prognostic genes. (**A**) Correlation between SPINK1 expression and the infiltration levels of six immune cell populations. (**B**) Correlation between TIMP1 expression and the infiltration levels of six immune cell populations.

**Figure 6 ijms-26-11964-f006:**
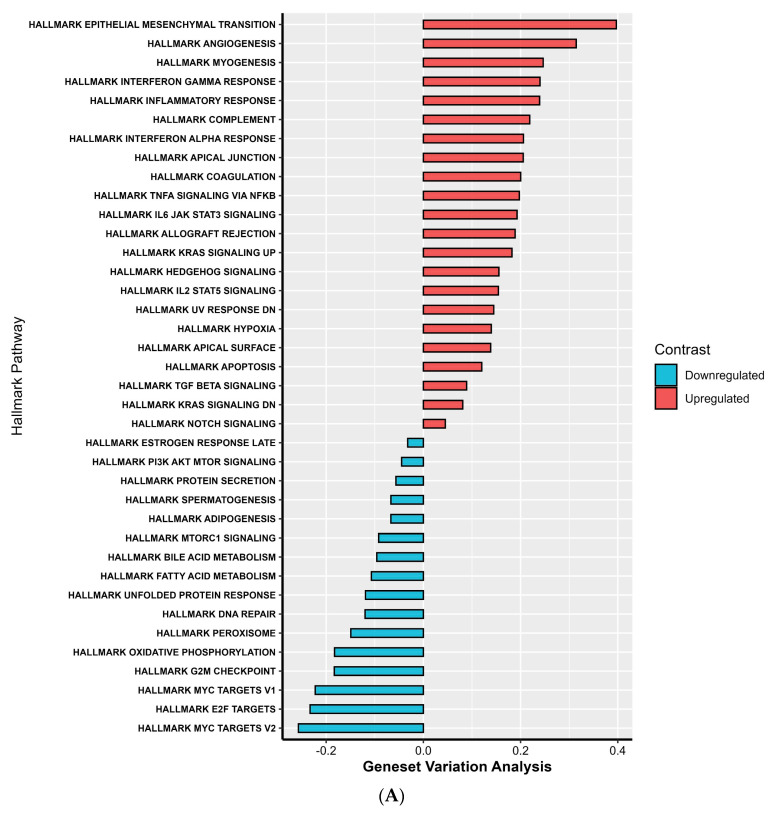
Gene Set Variation Analysis (GSVA) of hallmark and Reactome pathways between high- and low-risk groups. Differentially enriched pathways between high- and low-risk groups based on GSVA scores. Pathways with positive log fold change (logFC > 0) are upregulated in the high-risk group, while those with negative log fold change (logFC < 0) are upregulated in the low-risk group. (**A**) Hallmark pathways. (**B**) Reactome pathways.

**Table 1 ijms-26-11964-t001:** The clinicopathological characteristics across the risk groups.

Characteristic	High-Risk *n* = 187 ^1^	Low-Risk *n* = 188 ^1^	*p*-Value ^2^
Sex			0.6
Female	86 (46%)	82 (44%)	
Male	101 (54%)	106 (56%)	
Race			>0.9
American Indian	1 (0.6%)	0 (0%)	
Asian	7 (4.0%)	5 (3.0%)	
Black or African American	30 (17%)	30 (18%)	
White	138 (78%)	132 (79%)	
Ethnicity			0.6
Hispanic or Latino	3 (1.8%)	1 (0.6%)	
Not Hispanic or Latino	165 (98%)	158 (99%)	
Stage			0.7
Stage I	26 (14%)	31 (17%)	
Stage II	66 (36%)	69 (38.1%)	
Stage III	61 (34%)	52 (28%)	
Stage IV	25 (14%)	27 (15%)	
Vascular Invasion Indicator	49 (30%)	27 (17%)	0.007
Lymphovascular Invasion Indicator	61 (36%)	41 (25%)	0.031
Perineural Invasion	35 (29%)	23 (22%)	0.2
KRAS Mutation	17 (53%)	11 (46%)	0.6
BRAF Gene Analysis Abnormality			>0.9
Abnormal	1 (6.7%)	2 (10%)	
Normal	14 (93%)	18 (90%)	
Weight	76 (62, 90)	81 (69, 94)	0.037
Height	170 (162, 175)	170 (163, 179)	0.12
Age	66 (56, 74)	66 (55, 75)	0.8
Primary Site			0.002
Left-sided	80 (45%)	111 (62%)	
Right-sided	98 (55%)	69 (38%)	

^1^ *n* (%); Median (Q1, Q3); ^2^ Pearson’s Chi-squared test for categorical variables; Fisher’s exact test for categorical variables with counts < 5; Wilcoxon rank sum test for continuous variables.

**Table 2 ijms-26-11964-t002:** Immune infiltration analysis between the risk groups.

Characteristic	High-Risk *n* = 187 ^1^	Low-Risk *n* = 188 ^1^	*p*-Value ^2^
aDC	0.13 (0.06, 0.23)	0.05 (0.02, 0.10)	<0.001
Adipocytes	0.0000 (0.0000, 0.0005)	0.0000 (0.0000, 0.0002)	0.10
B-cells	0.022 (0.011, 0.046)	0.022 (0.014, 0.035)	0.7
Basophils	0.039 (0.021, 0.060)	0.037 (0.021, 0.051)	0.13
CD4+ memory T-cells	0.006 (0.003, 0.010)	0.009 (0.005, 0.012)	<0.001
CD4+ naive T-cells	0.015 (0.006, 0.025)	0.012 (0.004, 0.017)	0.001
CD4+ T-cells	0.003 (0.001, 0.007)	0.004 (0.002, 0.007)	0.3
CD4+ Tcm	0.016 (0.010, 0.025)	0.017 (0.011, 0.028)	0.6
CD4+ Tem	0.020 (0.009, 0.032)	0.020 (0.012, 0.033)	>0.9
CD8+ naive T-cells	0.004 (0.003, 0.006)	0.004 (0.002, 0.007)	>0.9
CD8+ T-cells	0.019 (0.012, 0.029)	0.020 (0.014, 0.026)	0.8
CD8+ Tcm	0.009 (0.004, 0.022)	0.007 (0.003, 0.014)	0.014
CD8+ Tem	0.004 (0.001, 0.010)	0.004 (0.001, 0.007)	0.10
cDC	0.027 (0.015, 0.047)	0.017 (0.008, 0.031)	<0.001
Class-switched memory B-cells	0.017 (0.010, 0.026)	0.014 (0.008, 0.023)	0.036
CLP	0.020 (0.009, 0.032)	0.024 (0.013, 0.036)	0.009
CMP	0.0002 (0.0000, 0.0012)	0.0001 (0.0000, 0.0010)	0.2
DC	0.007 (0.003, 0.015)	0.003 (0.001, 0.006)	<0.001
Endothelial cells	0.026 (0.012, 0.048)	0.013 (0.005, 0.023)	<0.001
Eosinophils	0.005 (0.003, 0.008)	0.003 (0.002, 0.006)	<0.001
Epithelial cells	0.060 (0.041, 0.075)	0.060 (0.048, 0.074)	0.5
Erythrocytes	0.0001 (0.0000, 0.0003)	0.0002 (0.0001, 0.0005)	<0.001
Fibroblasts	0.12 (0.07, 0.19)	0.06 (0.03, 0.11)	<0.001
GMP	0.011 (0.004, 0.017)	0.009 (0.004, 0.015)	0.4
iDC	0.05 (0.02, 0.11)	0.02 (0.01, 0.05)	<0.001
ly Endothelial cells	0.008 (0.004, 0.016)	0.004 (0.001, 0.006)	<0.001
Macrophages	0.015 (0.007, 0.030)	0.006 (0.003, 0.011)	<0.001
Macrophages M1	0.021 (0.010, 0.037)	0.008 (0.004, 0.015)	<0.001
Macrophages M2	0.010 (0.004, 0.017)	0.004 (0.002, 0.008)	<0.001
Mast cells	0.010 (0.006, 0.014)	0.009 (0.006, 0.012)	0.3
Memory B-cells	0.007 (0.003, 0.014)	0.008 (0.004, 0.015)	0.4
Monocytes	0.009 (0.003, 0.020)	0.003 (0.000, 0.008)	<0.001
MPP	0.0000 (0.0000, 0.0010)	0.0000 (0.0000, 0.0002)	0.6
MSC	0.17 (0.13, 0.23)	0.12 (0.08, 0.16)	<0.001
mv Endothelial cells	0.016 (0.009, 0.027)	0.008 (0.004, 0.014)	<0.001
naive B-cells	0.003 (0.001, 0.007)	0.004 (0.002, 0.007)	0.2
Neurons	0.0012 (0.0005, 0.0021)	0.0013 (0.0008, 0.0020)	0.4
Neutrophils	0.0005 (0.0000, 0.0017)	0.0003 (0.0000, 0.0016)	0.3
NK cells	0.0003 (0.0000, 0.0024)	0.0011 (0.0000, 0.0029)	0.023
NKT	0.022 (0.011, 0.040)	0.026 (0.013, 0.040)	0.13
pDC	0.003 (0.001, 0.007)	0.003 (0.001, 0.007)	0.2
Pericytes	0.09 (0.05, 0.13)	0.04 (0.02, 0.07)	<0.001
Plasma cells	0.007 (0.004, 0.011)	0.008 (0.006, 0.012)	0.015
Platelets	0.0040 (0.0025, 0.0066)	0.0042 (0.0028, 0.0070)	0.14
Preadipocytes	0.034 (0.017, 0.059)	0.024 (0.014, 0.038)	<0.001
pro B-cells	0.012 (0.005, 0.017)	0.013 (0.009, 0.019)	0.003
Tgd cells	0.012 (0.004, 0.018)	0.016 (0.009, 0.023)	<0.001
Th1 cells	0.08 (0.05, 0.12)	0.10 (0.07, 0.13)	<0.001
Th2 cells	0.07 (0.04, 0.12)	0.09 (0.05, 0.13)	0.018
Tregs	0.015 (0.009, 0.022)	0.014 (0.008, 0.019)	0.2
Immune Score	0.07 (0.04, 0.12)	0.05 (0.04, 0.07)	<0.001
Stroma Score	0.07 (0.05, 0.12)	0.03 (0.02, 0.06)	<0.001
Microenvironment Score	0.16 (0.10, 0.23)	0.10 (0.07, 0.12)	<0.001

^1^ Median (Q1, Q3); ^2^ Wilcoxon rank sum test; aDC: Activated Dendritic Cells; B-cells: B Lymphocytes; CD4+ Tcm: CD4-positive Central Memory T Cells; CD4+ Tem: CD4-positive Effector Memory T Cells; CD8+ Tcm: CD8-positive Central Memory T Cells; CD8+ Tem: CD8-positive Effector Memory T Cells; cDC: Conventional Dendritic Cells; CLP: Common Lymphoid Progenitor; CMP: Common Myeloid Progenitor; DC: Dendritic Cells; GMP: Granulocyte-Macrophage Progenitor; iDC: Immature Dendritic Cells; ly Endothelial cells: Lymphatic Endothelial Cells; MPP: Multipotent Progenitor; MSC: Mesenchymal Stem Cells; mv Endothelial cells: Microvascular Endothelial Cells; NK cells: Natural Killer Cells; NKT: Natural Killer T Cells; pDC: Plasmacytoid Dendritic Cells; Tgd cells: Gamma Delta T Cells; Th1 cells: T Helper 1 Cells; Th2 cells: T Helper 2 Cells; Tregs: Regulatory T Cells.

**Table 3 ijms-26-11964-t003:** Immune checkpoint expression and exhausted CD8+ T-cells markers between high- vs. low-risk groups.

Characteristic	High, *n* = 187 ^1^	Low, *n* = 188 ^1^	*p*-Value ^2^
CTLA4	5.27 (4.41, 6.06)	4.59 (3.67, 5.32)	<0.001
PD1	4.92 (3.88, 5.95)	4.13 (3.29, 4.94)	<0.001
PDL1	4.65 (3.68, 5.61)	3.71 (3.01, 4.73)	<0.001
LAG3	5.95 (5.05, 6.90)	5.04 (4.32, 5.82)	<0.001
TOX	6.92 (5.98, 8.05)	6.57 (5.58, 7.53)	0.11
EOMES	3.58 (2.33, 4.71)	2.63 (1.46, 3.52)	<0.001
TIGIT	5.82 (4.71, 6.71)	4.69 (3.67, 5.53)	<0.001
CTLA4 Group			<0.001
High	119 (64%)	69 (37%)	
Low	68 (36%)	119 (63%)	
PD1 Group			<0.001
High	121 (65%)	67 (36%)	
Low	66 (35%)	121 (64%)	
PDL1 Group			<0.001
High	118 (63%)	70 (37%)	
Low	69 (37%)	118 (63%)	
LAG3 Group			<0.001
High	118 (63%)	70 (37%)	
Low	69 (37%)	118 (63%)	
Exhausted CD8			<0.001
Exhausted CD8+	116 (62%)	59 (31%)	
No Exhausted CD8+	71 (38%)	129 (69%)	

^1^ Median (IQR); *n* (%); ^2^ Wilcoxon rank sum test; Pearson’s Chi-squared test.

## Data Availability

All data used in this study are publicly available on Gene Expression Omnibus (GEO) with accession number GSE39582, GSE17536 and The Cancer Genome Atlas (TCGA) TCGA-COAD. The data sources include publicly accessible databases and repositories. No new data was generated for this research.

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
