# Peer review of "Single-Cell and Bulk RNA Sequencing Reveal SPINK1 and TIMP1 as Epithelial Cell Marker Genes Linked to Colorectal Cancer Survival and Tumor Immune Microenvironment Profiles"

_ijms, 2025, doi:10.3390/ijms262411964_

Round 1
Reviewer 1 Report
Comments and Suggestions for Authors
The manuscript presents a combined single-cell and bulk RNA-seq analysis to identify epithelial marker genes (EMGs) associated with colorectal cancer prognosis, highlighting SPINK1 and TIMP1 as potential prognostic biomarkers. The study is scientifically interesting and touches on clinically relevant topics such as tumor–immune microenvironment, survival modeling, and multi-omics integration.
However, the manuscript suffers from significant methodological, structural, and clarity issues.
1 The novelty of the study is not clearly articulated. Several published works (e.g., https://doi.org/10.1002/cam4.4104, https://doi.org/10.1158/1538-7445.AM2025-5077 ) have already reported prognostic EMG signatures in CRC. Authors should clearly explain how their approach differs and why SPINK1/TIMP1 identification is novel.
The rationale for intersecting epithelial marker genes with malignant cell markers is insufficient. This step risks conceptual confusion: epithelial markers are typically non-malignant markers, while malignant markers represent transformed tissue. Authors must justify this intersection biologically.
The manuscript states that 20 clusters were detected but only 12 annotated cell types are shown. Please clarify how cluster merging or annotation was performed.
The criteria for selecting EMGs (|log2FC|>0.5, adj-p<0.05) are very permissive for scRNA-seq. No batch correction, doublet removal, or QC criteria are described. This creates substantial risk of false-positive markers.
Figure 2C is described as “interaction between epithelial cells and other cells,” but no computational method (CellChat? NicheNet? ligand-receptor scores?) is described. The figure appears to be a heatmap without an interaction algorithm explanation.
Survival analysis uses 1-month, 2-month, and 3-month survival probabilities, which are not clinically meaningful for CRC (patients survive years, not months). These survival intervals need justification or correction.
Stage, sex, and age are stated as non-significant in the univariate Cox model, but this contradicts known clinical evidence. Please check for overfitting, incorrect model coding, or missing covariates.
Using median-split risk groups is statistically outdated and can inflate significance. Consider time-dependent ROC, C-index, or restricted mean survival time (RMST).
The LASSO model workflow is inconsistent: text says 18 genes remained after Cox, 2 after LASSO, but figure 3A states “34 genes.” Please reconcile these inconsistencies.
Using TIMER and xCell together without adjusting for tumor purity creates confounders—especially TIMP1 is highly expressed by CAFs and ECM-rich tumors, biasing immune fractions.
Interpretation of dendritic cell (DC) infiltration is unclear. High DC infiltration does not necessarily indicate immune activation; in CRC, “tolerogenic DCs” are common and promote exhaustion.
The relationship between TIMP1 and CD8+ T-cell exhaustion is speculative in the discussion and not directly supported by the data. Exhaustion should be validated using multiple exhaustion markers (e.g., TOX, EOMES, TIGIT).
Hallmark and Reactome GSVA scores are reported but the gene sets used, software parameters, and p-adjustment method are not described.
The interpretation of mitochondrial tRNA processing pathway enrichment in low-risk patients is not biologically justified. Provide mechanistic rationale or remove speculation.
Only one external dataset (GSE17536) is used. GSE17536 alone is not sufficient to claim model robustness. Authors should consider including:
GSE39582
GSE14333
GSE38832
Several figures are too small to interpret (UMAPs, GSVA heatmaps). High-resolution versions are required.
Table 1 reports KRAS mutation rates incorrectly (values appear repeated for both groups). Please verify all clinical variables.
Grammar and writing quality should be improved across the manuscript (“we not statistically significant,” “CRC at single-cell resolution to identify epithelial marker genes”). A native English editing service is recommended.
The abbreviation EMG (epithelial marker gene) is used inconsistently. Sometimes refers to epithelial markers; other times to survival markers.
The Introduction is overly long and includes textbook information (risk factors, basic EMT). Consider shortening.
The Methods should include:
QC thresholds for scRNA data (min genes, max counts, mitochondrial %)
How missing clinical data were handled
Software versions for TIMER, xCell, GSVA, and Boruta
Figure 1 (workflow) appears after the Methods; move it earlier for clarity.
Author Response
Reviewer 1:
The manuscript presents a combined single-cell and bulk RNA-seq analysis to identify epithelial marker genes (EMGs) associated with colorectal cancer prognosis, highlighting SPINK1 and TIMP1 as potential prognostic biomarkers. The study is scientifically interesting and touches on clinically relevant topics such as tumor–immune microenvironment, survival modeling, and multi-omics integration.
However, the manuscript suffers from significant methodological, structural, and clarity issues.
- The novelty of the study is not clearly articulated. Several published works (e.g., https://doi.org/10.1002/cam4.4104, https://doi.org/10.1158/1538-7445.AM2025-5077 ) have already reported prognostic EMG signatures in CRC. Authors should clearly explain how their approach differs and why SPINK1/TIMP1 identification is novel.
- Thank you for your comment. First, unlike the cited work (https://doi.org/10.1002/cam4.4104), which identifies prognostic biomarkers exclusively from bulk RNA-sequencing data and is not restricted to epithelial-specific signatures, our study integrates both single-cell RNA-seq and bulk RNA-seq analyses. This combined strategy provides higher cellular resolution and allows us to map EMG dynamics specifically to epithelial cell states, thereby reducing noise introduced by stromal or immune components in bulk-only approaches. Second, the second reference provided is a conference abstract corresponding to this full-text publication. Importantly, our study is the first to identify SPINK1 and TIMP1 as epithelial-specific EMG signatures using integrated scRNA + bulk data, and to validate their prognostic relevance through a multi-step pipeline. We have added a paragraph in the Discussion section to clearly highlight these distinctions and contextualize our contributions relative to existing literature.
- The rationale for intersecting epithelial marker genes with malignant cell markers is insufficient. This step risks conceptual confusion: epithelial markers are typically non-malignant markers, while malignant markers represent transformed tissue. Authors must justify this intersection biologically.
- Thank you for your comment. The intersection step was designed to identify the epithelial cell origin genes that become dysregulated during malignant transformation. By the intersection, we aimed to capture cell-type-specific drivers of tumorigenesis, rather than non-specific markers expressed across multiple cell types.
- The manuscript states that 20 clusters were detected but only 12 annotated cell types are shown. Please clarify how cluster merging or annotation was performed.
- Thank you for your comment. The 20 clusters presented in the UMAP reflect results of the unsupervised clustering from the TISCH/MAESTRO pipeline, where the transcriptionally distinct subpopulations are detected without prior biological knowledge. Following this, the TISCH assigns biological identities to clusters using a hierarchical annotation strategy. For malignant clusters, annotation combines the annotations provided by the original study, the expression of malignant markers, and the copy number variation inference using InferCNV. For the non-malignant clusters, the annotation pipeline assigns clusters to cell types by comparing cluster-specific differentially expressed genes to curated marker gene sets, followed by manual curation based on the original study’s annotations. Several clusters were annotated as belonging to the same lineage, and thus lineage-level grouping reduced the 20 transcriptional clusters to 12 biologically interpretable cell types. We added a clarification in the methods section. We clarified this in the results section.
- The criteria for selecting EMGs (|log2FC|>0.5, adj-p<0.05) are very permissive for scRNA-seq. No batch correction, doublet removal, or QC criteria are described. This creates a substantial risk of false-positive markers.
- Thank you for your comment. The epithelial-marker genes (EMGs) were extracted from the TISCH2 database, which uniformly processes all tumor scRNA-seq datasets using the MAESTRO pipeline. As documented by TISCH2, MAESTRO performs systematic quality control (including doublet and low-quality cell removal), normalization, clustering, and curated cell-type annotation before any differential expression analysis is generated. After the preprocessing, TISCH further assigns cell identities at three hierarchical levels (malignancy, major lineage, and minor lineage), ensuring that epithelial markers arise from rigorously curated epithelial clusters rather than from noise and unfiltered cells. As this pipeline already applies stringent QC and statistical modeling upstream, our additional cutoff (|log2FC| > 0.5 and adjusted p < 0.05) serves only as a secondary filtering step to retain robust EMGs, rather than defining them de novo from raw data. We clarified this in the methods section.
- Figure 2C is described as “interaction between epithelial cells and other cells,” but no computational method (CellChat? NicheNet? ligand-receptor scores?) is described. The figure appears to be a heatmap without an interaction algorithm explanation.
- Thank you for your comment. In Figure 2C, the interaction heatmap and network visualization were generated from pre-computed cell–cell communication analysis available in TISCH database, which integrates the CellChat algorithm to infer ligand–receptor-based signaling between cell clusters. CellChat quantifies the number of significant ligand–receptor pairs between any two cell populations using its statistical framework, and the number of significant interactions is used as the interaction score. Hence, the heatmap in the figure represents the total number of statistically significant ligand–receptor interactions between each pair of clusters, whereas the network diagram shows the interaction network of the selected epithelial cluster, with edge width proportional to the number of significant ligand–receptor pairs. We have updated the Results section to clarify this process.
- Survival analysis uses 1-month, 2-month, and 3-month survival probabilities, which are not clinically meaningful for CRC (patients survive years, not months). These survival intervals need justification or correction.
- Thank you for your comment. We have updated our analysis to assess survival probabilities at 1-year, 3-year, and 5-year.
- Stage, sex, and age are stated as non-significant in the univariate Cox model, but this contradicts known clinical evidence. Please check for overfitting, incorrect model coding, or missing covariates.
- Thank you for your comment. Upon re-evaluation, age was found to be significantly associated with survival in the univariable Cox analysis, alongside SPINK1 and TIMP1. Sex and tumor stage remained non-significant in our cohort. The initial analysis did not detect age as significant because missing clinical data in these variables were not fully addressed previously. The results and corresponding text in the manuscript have been updated to reflect these findings.
- Using median-split risk groups is statistically outdated and can inflate significance. Consider time-dependent ROC, C-index, or restricted mean survival time (RMST).
- Thank you for your comment. We added a complementary analysis of the continuous risk score, including time-dependent ROC, C-index, and restricted mean survival time (RMST), which confirms the prognostic value of the risk score. The manuscript has been updated to include these metrics in the Results section.
- The LASSO model workflow is inconsistent: text says 18 genes remained after Cox, 2 after LASSO, but figure 3A states “34 genes.” Please reconcile these inconsistencies.
- Thank you for pointing out. Fixed and updated.
- Using TIMER and xCell together without adjusting for tumor purity creates confounders—especially TIMP1 is highly expressed by CAFs and ECM-rich tumors, biasing immune fractions.
- Thank you for your comment. We acknowledge that using TIMER and xCell without tumor purity adjustment may introduce biases. In the revised manuscript, we have added this point as a limitation and clarified that single-cell or purity-adjusted deconvolution will be required to validate immune associations in future studies.
- Interpretation of dendritic cell (DC) infiltration is unclear. High DC infiltration does not necessarily indicate immune activation; in CRC, “tolerogenic DCs” are common and promote exhaustion.
- Thank you for your comment. We agree with your point. We have revised the Discussion to remove overinterpretation and now describe this as a potential immunosuppressive feature that warrants further validation.
- The relationship between TIMP1 and CD8+ T-cell exhaustion is speculative in the discussion and not directly supported by the data. Exhaustion should be validated using multiple exhaustion markers (e.g., TOX, EOMES, TIGIT).
- Thank you for pointing out. We inferred the presence of exhausted CD8+ T-cells by grouping patients expressing the following checkpoint levels: High PD1 + High LAG3, High PD1 + High CTLA-4, and High PD1 + High PDL-1. We further analyzed the expression of exhaustion markers (TOX, EOMES, TIGIT) in the revised manuscript (Table 3).
- Hallmark and Reactome GSVA scores are reported but the gene sets used, software parameters, and p-adjustment method are not described.
- Thank you for pointing out. Added.
- The interpretation of mitochondrial tRNA processing pathway enrichment in low-risk patients is not biologically justified. Provide mechanistic rationale or remove speculation.
- Thank you for pointing out. We observed that the mitochondrial tRNA processing pathway was enriched in the low-risk group, but we did not make any speculation on it as the biological relevance is uncertain.
- Only one external dataset (GSE17536) is used. GSE17536 alone is not sufficient to claim model robustness. Authors should consider including: GSE39582, GSE14333, GSE38832.
- Thank you for your suggestions. In the revised version, we add the validation cohort GSE39582, which includes a substantially larger number of patients and provides greater statistical power for validating the prognostic model.
- Several figures are too small to interpret (UMAPs, GSVA heatmaps). High-resolution versions are required.
- Thank you for your comment. A separate compressed folder for the figures with higher resolution was uploaded.
- Table 1 reports KRAS mutation rates incorrectly (values appear repeated for both groups). Please verify all clinical variables.
- Thank you for pointing out. Fixed.
- Grammar and writing quality should be improved across the manuscript (“we not statistically significant,” “CRC at single-cell resolution to identify epithelial marker genes”). A native English editing service is recommended.
- Thank you for your suggestion. Revised and checked.
- The abbreviation EMG (epithelial marker gene) is used inconsistently. Sometimes refers to epithelial markers; other times to survival markers.
- Thank you for pointing out. Adjusted.
- The Introduction is overly long and includes textbook information (risk factors, basic EMT). Consider shortening.
- Thank you for pointing out. We have shortened the introduction to focus on key background and rationale for the study.
- The Methods should include:
- QC thresholds for scRNA data (min genes, max counts, mitochondrial %)
- Thank you for pointing out. As the scRNA-seq data were processed using TISCH, standard QC thresholds such as minimum genes per cell, maximum counts, and mitochondrial percentage were addressed during the preprocessing.
- How missing clinical data were handled.
- Thank you for pointing out. Patients with incomplete survival outcomes or missing clinical data were excluded from the analysis.
- Software versions for TIMER, xCell, GSVA, and Boruta
- Thank you for pointing out. Added.
- Figure 1 (workflow) appears after the Methods; move it earlier for clarity.
- Thank you for pointing out. Adjusted.

Reviewer 2 Report
Comments and Suggestions for Authors
Please see the uploaded atttachment.

Author Response
Reviewer 2:
- Lack of mechanistic or causal explanation linking the identified biomarkers to patient survival. Although the study identifies SPINK1 and TIMP1 as independent prognostic factors, the evidence presented is purely statistical and does not clarify the causal mechanisms through which these genes influence survival in colorectal cancer. This represents a major limitation of the current work. I recommend that the authors explicitly acknowledge this issue in the Discussion section. Moreover, future studies could follow the approach of Zhang et al. (PMID: 39680087) to incorporate experimental validation or causal inference frameworks to elucidate the biological functions of the proposed biomarkers.
- Thank you for your comment. We revised the discussion accordingly to add on the biological significance and how they influence survival in colorectal cancer.
- The prognostic model is constructed using single-gene–based features rather than gene-pair–based approaches. The proposed risk score is derived from two individual genes (SPINK1 and TIMP1), which is essentially a traditional single-gene modeling strategy. However, prior research has demonstrated that gene-pair–based signatures may provide superior robustness and predictive performance. For instance, Xie et al. (PMID: 35143414) showed that gene-pair–based models outperform single-gene features by mitigating platform heterogeneity, normalization biases, and batch-related variability. The authors should acknowledge this limitation and discuss how future work might incorporate gene-pair frameworks or compare their model with existing gene-pair signatures to enhance rigor and generalizability
- Thank you for pointing out. We added to the limitations section according to your comment.

Round 2
Reviewer 1 Report
Comments and Suggestions for Authors
The manuscript presents valuable findings and is suitable for publication after minor revision. However, the text currently contains numerous grammatical and language-related issues that significantly affect readability and scientific clarity. A thorough English language editing by a professional or a native speaker is strongly required before the manuscript can be accepted for publication.
Additionally, the figures in the current version are inferior in quality. The authors should replace all low-resolution images with high-quality versions that meet the journal’s standards, particularly for the online publication.
Once these issues are correctly addressed, the manuscript will be acceptable for publication.
Author Response
The manuscript presents valuable findings and is suitable for publication after minor revision. However, the text currently contains numerous grammatical and language-related issues that significantly affect readability and scientific clarity. A thorough English language editing by a professional or a native speaker is strongly required before the manuscript can be accepted for publication.
Additionally, the figures in the current version are inferior in quality. The authors should replace all low-resolution images with high-quality versions that meet the journal’s standards, particularly for the online publication.
Once these issues are correctly addressed, the manuscript will be acceptable for publication.
- Additional language modifications and grammar corrections were performed.
Reviewer 2 Report
Comments and Suggestions for Authors
Congratulations!
Author Response
Congratulations!
Thank you!